# Update on Targeted Therapy and Immunotherapy for Metastatic Colorectal Cancer

**DOI:** 10.3390/cells13030245

**Published:** 2024-01-28

**Authors:** Patrick W. Underwood, Samantha M. Ruff, Timothy M. Pawlik

**Affiliations:** Department of Surgery, The Ohio State University Wexner Medical Center and James Comprehensive Cancer Center, 395 W. 12th Ave., Suite 670, Columbus, OH 43210, USA; patrick.underwood@osumc.edu (P.W.U.); samantha.ruff@osumc.edu (S.M.R.)

**Keywords:** targeted therapy, personalized medicine, colorectal cancer, molecular profiling

## Abstract

Metastatic colorectal cancer remains a deadly malignancy and is the third leading cause of cancer-related death. The mainstay of treatment for metastatic colorectal cancer is chemotherapy, but unfortunately, even with recent progress, overall survival is still poor. Colorectal cancer is a heterogeneous disease, and the underlying genetic differences among tumors can define the behavior and prognosis of the disease. Given the limitations of cytotoxic chemotherapy, research has focused on developing targeted therapy based on molecular subtyping. Since the early 2000s, multiple targeted therapies have demonstrated efficacy in treating metastatic colorectal cancer and have received FDA approval. The epidermal growth factor receptor (EGFR), vascular endothelial growth factor (VEGF), and DNA mismatch repair pathways have demonstrated promising results for targeted therapies. As new gene mutations and proteins involved in the oncogenesis of metastatic colorectal cancer are identified, new targets will continue to emerge. We herein provide a summary of the updated literature regarding targeted therapies for patients with mCRC.

## 1. Introduction

Colorectal cancer (CRC) is the third most commonly diagnosed cancer in men and women in the United States and the third leading cause of cancer-related death [1]. Data from the Surveillance, Epidemiology, and End Results (SEER) database demonstrates that the incidence of CRC has decreased between 1985 and 2019 [1]. This trend is likely due to improvement in screening modalities with colonoscopy and stool tests [2]. Unfortunately, 20% of patients will still present with synchronous metastases, and another 40% who present with locoregional disease will develop metastatic disease, depending on differences in the underlying biology of different subtypes of CRC [3,4]. In addition, suboptimal screening and surveillance of CRC may be due to socioeconomic disparities [2,5].

Patients with metastatic CRC (mCRC) should be managed by a multi-disciplinary team and be treated with a combination of surgery, systemic therapy, and/or locoregional therapy (e.g., radiation therapy, hepatic artery infusion pump). Systemic therapy represents the mainstay of treatment and for some patients with advanced disease, it is the best therapeutic option. Despite the presence of effective cytotoxic chemotherapies, five-year survival for patients with advanced disease is low (10–30%) [4,6,7].

Underlying genetic differences among tumors can define the behavior and prognosis of the disease. Given the limitations of cytotoxic chemotherapy, research has focused on developing targeted therapy based on molecular subtyping. Over the last two decades, a growing number of targeted therapies have been demonstrated to be effective in producing tumor regression and improved survival. Additionally, targeted therapies directed at the biologic features of cancers (e.g., angiogenesis) may have less effect on healthy cells and a better safety profile versus traditional cytotoxic chemotherapy. We herein review and summarize the current literature related to targeted therapies for patients with mCRC.

## 2. Methods

A broad review of the literature was performed using MEDLINE/PubMed. The search date ended on 15 December 2023. The terms “targeted therapy”, “colon cancer”, colorectal cancer”, and “immunotherapy” were searched. The search results were reviewed by the co-authors. The inclusion of studies focused on clinical trials related to targeted therapy and immunotherapy for the treatment of colorectal cancer metastasis was prioritized. Two authors (P.U. and S.R) performed the initial review of the literature, and final determinations about article inclusion were made by the senior author (T.P.).

## 3. Chemotherapy, Targeted Therapy, and Molecular Profiling

There are multiple effective systemic chemotherapy regimens available for use in colorectal cancer. Common regimens are either oxaliplatin-based (FOLFOX or CAPOX) or irinotecan-based (FOLFIRI or CAPIRI). The National Comprehensive Cancer Network (NCCN) guidelines recommend the use of doublet or triplet therapy for patients who can tolerate an intensive therapeutic regimen. Multiple studies have demonstrated similar survival for different combination regimens but different toxicity profiles [8,9,10]. Targeted therapies improve response to therapy and survival when combined with traditional systemic chemotherapy. In contrast, for patients who cannot tolerate an intensive regimen, single-agent therapy may be used (5-FU ± leucovorin or capecitabine).

Cytotoxic chemotherapy kills cells as they replicate without differentiating between malignant and healthy tissue. In contrast, targeted therapies work on cancer cells by directly targeting proteins or cells involved in cell proliferation, growth, and metastasis. There are many potential therapeutic targets for patients with mCRC. For targeted therapies to be effective, the relevant proteins or genetic mutations must be present. This makes molecular profiling of tumors critically important. The EGFR, RAS, BRAF, VEGF, and HER2 pathways have emerged as important targets [11]. Investigations into numerous other potential targets are ongoing [12]. Table 1 displays the current Food and Drug Administration (FDA)-approved targeted therapies for mCRC.

## 4. The Epidermal Growth Factor Receptor Pathway

The epidermal growth factor receptor (EGFR) is a transmembrane protein that can be bound by specific ligands, including epidermal growth factor and transforming growth factor alpha (Figure 1). The EGFR belongs to the ErbB family of four structurally related receptor tyrosine kinases. After ligand binding, dimerization occurs, which activates a downstream pathway of signaling proteins, including RAS/RAF/MEK/ERK, PI3K/AKT, and JAK/STAT3. These proteins are involved in cell growth and proliferation. Overexpression can lead to carcinogenesis. There are multiple proteins in the EGFR pathway that have been targeted for the treatment of CRC.

### 4.1. Cetuximab

Cetuximab is a chimeric (mouse/human) monoclonal IgG1 antibody developed to bind the extracellular EGFR domain that blocks dimerization and ligand-induced signaling of the EGFR pathway [14]. Cetuximab was first approved by the FDA in 2004 after a multicenter, randomized trial in Europe demonstrated promising results among patients with poor response to irinotecan-based regimens [15]. A randomized controlled trial published by Jonker et al. evaluated the use of cetuximab plus best supportive care versus best supportive care alone in patients with colorectal cancer [14]. Patients who received cetuximab had a 6.1-month median survival compared with 4.6 months among individuals in the best supportive care group. 

More recent trials have evaluated the use of cetuximab as an adjunct to traditional chemotherapy regimens. The CRYSTAL trial was a multicenter, phase 3 trial evaluating FOLFIRI plus cetuximab versus FOLFIRI alone for patients with unresectable colorectal cancer [16]. The results of this trial demonstrated a decrease in progression-free survival but no difference in overall survival. Sub-analysis revealed that the benefits of cetuximab were limited to patients with KRAS wild-type (wt) tumors. These data highlighted the importance of KRAS mutation testing in patient treatment and clinical trial stratification. A more recent multicenter, randomized trial in China, the TAILOR trial, compared FOLFOX with or without cetuximab in patients with KRAS wt tumors [17]. This trial demonstrated improved progression-free survival and overall survival time (20.7 vs. 17.8 months) in patients treated with FOLFOX and cetuximab. Interestingly, there does appear to be one subgroup of KRAS-mutated tumors that benefit from cetuximab therapy. Specifically, an analysis of multiple clinical trials revealed that patients with KRAS G13D mutations treated with cetuximab had better overall and progression-free survival versus other KRAS mutations [18]. A similar analysis evaluating panitumumab did not demonstrate improved survival for any KRAS mutation subgroups [19]. The results of this study support the use of panitumumab for only KRAS wt tumors.

Cetuximab plus chemotherapy is commonly used for patients with mCRC. The results of the recent EPOC trial serve to caution its use in patients with operable liver metastases [20]. This study was a phase 3 multicenter, randomized trial comparing chemotherapy with or without cetuximab before and after liver resection. Interestingly, the median overall survival was 81.0 months in the chemotherapy group and 55.4 months in the cetuximab plus chemotherapy group. With the large difference in median overall survival, the authors concluded that cetuximab should not be used in the operable setting. 

### 4.2. Panitumumab

Panitumumab was developed as an alternative humanized monoclonal antibody to the EGFR [21]. Similar to cetuximab, panitumumab binds the EGFR to prevent its activation. Patients must have KRAS wt tumors for panitumumab to be effective [22,23]. The addition of panitumumab to the best supportive care demonstrated improvement in progression-free survival in patients who had progressed on first-line therapy [24]. Recent studies have evaluated the addition of panitumumab to other chemotherapy regimens. A phase III randomized trial demonstrated that panitumumab added to FOLFIRI was superior to FOLFIRI alone for progression-free survival in patients with progression on one prior chemotherapy regimen [23]. A trend toward improved overall survival was noted in this study in the FOLFIRI/panitumumab cohort but was not statistically significant (14.5 versus 12.5 months, respectively, *p* = 0.12). The PRIME trial evaluated FOLFOX with or without panitumumab in patients with KRAS wt mCRC as first-line therapy [25]. Similar to cetuximab, this multinational, multicenter, randomized trial demonstrated improved progression-free survival with the addition of panitumumab, but not overall survival. 

### 4.3. Cetuximab vs. Panitumumab

Similar mechanisms of action and findings in clinical trials have led to investigations directly comparing cetuximab and panitumumab. As noted above, mutations to the RAS (KRAS and NRAS) pathway can affect response to EGFR-targeted therapies [26,27]. Subsequent trials focused on patients with KRAS and NRAS wt tumors. The ASPECCT trial was a multicenter, randomized trial comparing best supportive care with either cetuximab or panitumumab for patients with KRAS wt mCRC refractory to chemotherapy [28]. Of note, panitumumab was non-inferior to cetuximab. The median overall survival for the panitumumab group was 10.4 months and 10.0 months in the cetuximab group. Grade 3–4 toxicities were also similar between the groups. The WJOG 6510G trial compared irinotecan plus panitumumab or cetuximab in patients with KRAS wt mCRC and previous treatment with 5-Fu, oxaliplatin, and irinotecan-based therapies [29]. Again, panitumumab was non-inferior to cetuximab relative to progression-free survival and overall survival. Interestingly, median overall survival was 14.85 months in the panitumumab groups versus 11.53 months in the cetuximab group (*p* = 0.05).

### 4.4. Right- versus Left-Sided Colon Cancer

The embryologic origins of the right and left colon are different. The right colon derives its blood supply from the superior mesenteric artery and develops with the midgut while the left colon derives its blood supply from the inferior mesenteric artery and develops with the hindgut. This fact has led scientists to question whether right- and left-sided colon cancers may be biologically distinct with different survival and response to treatment options [30]. Right-sided tumors more commonly exhibit DNA mismatch repair deficiencies, while left-sided tumors often have mutations in KRAS, APC, PIK3CA, and p53 [30]. Right-sided cancers are associated with better overall prognosis at early stages and left-sided cancers are associated with better overall prognosis at late stages [31,32]. 

A meta-analysis of the CRYSTAL, PRIME, TAILOR, and 20,050,181 trials evaluated chemotherapy with or without cetuximab or panitumumab [33]. Cetuximab or panitumumab, in addition to chemotherapy, improved overall survival in left-sided but not right-sided colon cancer. There was, however, improved progression-free survival and objective response rates in both groups. Other studies have provided similar results [34,35]. Based on these analyses, current guidelines recommend cetuximab plus chemotherapy for left-sided KRAS wt tumors.

### 4.5. BRAF and MEK Inhibition

BRAF is a proto-oncogene that encodes a protein called B-RAF. The BRAF protein is downstream in the EGFR pathway. BRAF V600E mutations have been implicated in a number of malignancies. Among patients with colorectal cancer, there is an approximately 10% incidence of BRAF mutation [36]. MEK is another kinase in the EGFR pathway with potential as a target. Multiple therapies have been developed to target these pathways. Encorafenib is a BRAF inhibitor. Binimetinib is an MEK inhibitor. Pre-clinical studies and early trials have demonstrated improved responses to combined MEK and BRAF inhibition [37,38]. These data led to the BEACON trial, a phase 3 trial comparing three different groups of patients (i.e., encorafenib, binimetinib, and cetuximab; encorafenib and cetuximab; or control) with BRAF V600E mutations [39]. The control group was the investigator’s choice of cetuximab and irinotecan or cetuximab and FOLFIRI. This trial demonstrated improved survival in the triplet therapy group versus the doublet and control groups. Specifically, median overall survival was longer in the triplet therapy group (9.0 months) versus the other groups (5.4 months), as was the response rate and progression-free survival.

### 4.6. RAS Inhibition

KRAS represents another potentially targetable mutation in the EGFR pathway. EGFR activation leads to the activation of KRAS. KRAS mutations have been implicated in many malignancies. There are several different KRAS mutations that can occur. As noted above, KRAS mutant tumors do not respond well to EGFR inhibitors. KRAS G12C mutation occurs in about 4% of colorectal cancers [40]. Two therapies targeting KRAS G12C mutation are currently under investigation. Sotorasib irreversibly binds the KRAS G12C protein. A recent phase 3, multicenter, randomized trial compared sotorasib plus panitumumab to standard care in patients with mCRC refractory to standard therapy [41]. There were two sotorasib plus panitumumab groups with doses of sotorasib of 960 or 240 mg daily. The standard of care was the investigator’s choice of trifluridine–tipiracil or regorafenib. While overall survival data are not yet available, the median progression-free survival was longer in the 960 mg sotorasib/panitumumab group at 5.6 months versus 3.9 months in the 240 mg sotorasib/panitumumab group and 2.2 months in the standard therapy group. Toxicity profiles were similar between groups. Adagrasib is another KRAS G12C protein inhibitor. A phase 1–2, nonrandomized trial evaluated the safety of adagrasib or adagrasib in combination with cetuximab [42]. The dual therapy group had response rates of 46%, response duration of 7.6 months, and median progression-free survival of 6.9 months. A phase 3 clinical trial is currently recruiting patients.

### 4.7. Summary

The EGFR pathway offers multiple targets for therapy in patients with mCRC, including EGFR, BRAF, and RAS. Cetuximab and panitumumab have demonstrated a survival benefit in left-sided KRAS wt tumors. In turn, the NCCN guidelines recommend the use of these agents in this setting. Additionally, encorafenib is recommended for patients with BRAF V600E mutations. For patients with KRAS C12C mutations, guidelines recommend the use of sotorasib or adagrasib.

## 5. The Vascular Endothelial Growth Factor Pathway

The vascular endothelial growth factor (VEGF) pathway is another commonly targeted pathway for mCRC (Figure 2). The VEGF is a growth factor that stimulates the formation of blood vessels. Solid tumors require a blood supply to allow for cell growth and proliferation. VEGF signaling has been implicated in a number of solid tumors. These tumors hijack the pathway to promote angiogenesis and subsequent proliferation. The mechanisms through which this occurs are not well understood [43]. VEGF signaling is upregulated in CRC. Blockage of the pathway is, therefore, a potential target for therapy.

### 5.1. Bevacizumab

Bevacizumab is a humanized monoclonal antibody that targets the VEGF pathway. It acts by binding and circulating the VEGF and preventing binding to the VEGF receptor, which prevents angiogenesis [45]. It was the first anti-VEGF therapy approved by the FDA in 2004. Initial data from a multicenter, randomized trial compared irinotecan, bolus fluorouracil, and leucovorin (IFL) with or without bevacizumab in patients with previously untreated mCRC [46]. The median overall survival in the IFL and bevacizumab arm was 20.3 months versus 15.6 months in the IFL alone arm. Recent studies have evaluated the use of bevacizumab with more modern chemotherapy regimens. A multicenter, randomized trial compared FOLFOX4 plus bevacizumab, FOLFOX4 alone, and bevacizumab alone as second-line therapy for patients who already received fluoropyrimidine and irinotecan [47]. There was improved overall survival and progression-free survival in the FOLFOX4 plus bevacizumab compared with the other two groups. Another multicenter, randomized trial used a 2 × 2 factorial design to assign patients to XELOX versus FOLFOX4 with bevacizumab or placebo [48]. The oxaliplatin-based regimen with bevacizumab demonstrated improved progression-free survival (9.4 versus 8.0 months). 

A multinational, multicenter, randomized trial carried out in Europe compared a new chemotherapy regimen with or without bevacizumab for patients treated previously with a different chemotherapy regimen with bevacizumab [49]. The type of chemotherapy depended on the first line of treatment from which the patient was switched. There was an increase in median overall survival from 9.8 months to 11.2 months in the group that continued bevacizumab compared with the placebo group. Bevacizumab has also been examined in a randomized trial as maintenance therapy during chemotherapy-free intervals [50]. This trial demonstrated no benefit in overall survival or progression-free survival for bevacizumab maintenance therapy. Another trial evaluated the use of bevacizumab as adjuvant therapy after surgery [51]. This multicenter, randomized trial noted that bevacizumab plus FOLFOX4 did not improve survival compared with FOLFOX4 alone and appeared to have a detrimental effect on overall survival. Similar to cetuximab, bevacizumab is not recommended in the adjuvant setting. 

### 5.2. Aflibercept

Afilbercept, also known as Ziv-aflibercept, was developed as an alternative therapy targeting the VEGF pathway. The mechanism of action is described as a VEGF trap, which involves the Fc portion of human IgG fused to VEGF-binding portions of the VEGF receptor. Afilbercept binds to VEGF ligands to prevent binding to endogenous receptors [52]. Afilbercept received FDA approval in 2012 after clinical trial data, supporting its use in mCRC. The VELOUR trial, a multicenter, randomized trial, assigned patients with mCRC to FOLFIRI with or without aflibercept in patients as second-line therapy [53]. Patients assigned to the aflibercept group had improved overall survival (13.5 versus 12.1 months) and progression-free survival (6.9 versus 4.7 months). Studies are ongoing to compare aflibercept to bevacizumab, but no data are available at this time [54]. 

### 5.3. Ramucirumab

Ramucirumab is another anti-angiogenic agent developed to target the VEGF pathway. Ramucirumab is a human IgG monoclonal antibody that targets the VEGF2 receptor. The RAISE trial was a multicenter, randomized phase 3 trial that evaluated FOLFIRI with and without ramucirumab as second-line therapy in patients with metastatic colorectal cancer [55]. Median overall survival was higher in the FOLFIRI with ramucirumab group versus FOLOFIRI with placebo (13.3 months versus 11.7 months, respectively). A post hoc analysis of the RAISE trial demonstrated that the greatest benefit of ramucirumab therapy was in patients with CEA ≤ 10 versus CEA > 10 [56].

### 5.4. Regorafenib

Regorafenib is an anti-angiogenic multi-kinase inhibitor with action against multiple protein kinases. It functions through a blockade of VEGF and TIE2, leading to anti-angiogenic effects [57]. The utility of regorafenib has been studied in several different solid tumors and has been evaluated in a multinational, multicenter randomized, controlled trial [58]. The CORRECT trial tested regorafenib versus placebo in patients who had progressed on standard therapy. In this trial, 760 patients received regorafenib, while 753 received placebo. Median overall survival was improved in the regorafenib group compared with placebo (6.4 months versus 5.0 months). The CONCUR trial demonstrated similar results in an Asian population [59]. This multicenter, randomized trial compared regorafenib plus best supportive care to placebo plus best supportive care. Survival was 8.8 months in the regorafenib group versus 6.3 months in the placebo group. In addition, the CONSIGN study was a Phase IIIB study that evaluated the safety profile of regorafenib and demonstrated a reasonable toxicity profile for most patients [60].

### 5.5. Fruquitinib

Fruquitinib is the most recent FDA-approved therapy to target the VEGF pathway in metastatic colorectal cancer. Fruquitinib is a kinase inhibitor targeting VEGF receptors, which was evaluated in the FRESCO trial [61]. This multicenter, randomized trial in China compared fruquitinib to placebo in patients who had progressed on at least two lines of therapy but had not received a VEGF inhibitor. The study demonstrated improved median overall survival with fruquitinib versus placebo (9.3 months versus 6.6 months, respectively). Median progression-free survival was also improved (3.7 months versus 1.8 months). The FRESCO-2 trial was a multinational, multicenter, randomized trial that compared fruquitinib versus placebo in patients with the treatment of refractory mCRC [62]. Median overall survival was 7.4 months in the fruquitinib group versus 4.8 months in the placebo group (*p* < 0.001). The results of these studies resulted in recent FDA approval for the treatment of refractory mCRC.

### 5.6. Summary

Multiple drugs targeted at the VEGF receptor have been developed. Bevacizumab has been available for nearly two decades. Bevacizumab is often added to chemotherapy regimens to treat metastatic colorectal cancer and is recommended by NCCN guidelines. Alternative agents include aflibercept and ramucirumab. Fruquitinib and regorafenib are available and recommended for use by NCCN guidelines for patients who have failed trials of other therapies.

## 6. Comparing the EGFR and VEGF Pathways

As multiple targeted therapies have demonstrated improvement in outcomes for patients with mCRC, other trials have investigated outcomes among therapies that target different pathways. For example, the FIRE-3 trial compared two therapies first approved by the FDA [63]. Specifically, this multicenter, randomized trial performed in Germany and Austria compared FOLFIRI plus cetuximab to FOLFIRI plus bevacizumab in patients with KRAS wt tumors as first-line therapy. The trial enrolled 592 patients. The two groups demonstrated similar objective response rates and median progression -ree survival. Overall survival was higher in the cetuximab group (28.7 months versus 25.0 months). Post hoc radiologic analysis of the trial population demonstrated that tumor response and extent of response were correlated with the overall survival benefit [64].

A multicenter, randomized trial in the United States and Canada enrolled 1137 patients to compare cetuximab versus bevacizumab in combination with either mFOLFOX6 or FOLFIRI as first-line therapy [65]. Only patients with KRAS wt tumors were enrolled. This trial demonstrated that patients in the cetuximab plus chemotherapy group had slightly higher overall survival versus patients in the bevacizumab plus chemotherapy group (30.0 months versus 29.0 months). Similar to the FIRE-3 trial, median progression-free survival and response rates were similar between groups. A third multicenter, randomized trial in Japan compared mFOLFOX6 plus panitumumab or bevacizumab [66]. The trial enrolled 823 patients. Patients in the panitumumab group had longer median overall survival versus patients in the bevacizumab group (36.2 versus 31.3 months, respectively). In patients with left-sided tumors, median overall survival was 37.9 months in the panitumumab group and 34.3 months in the bevacizumab group. This trial also demonstrated improved response rates in the panitumumab group.

A meta-analysis of 13 trials evaluated the use of anti-EGFR and anti-VEGF treatments in patients with metastatic KRAS wt CRC based on primary tumor location [67]. There was improved survival in patients with left-sided tumors treated with anti-EGFR treatment. In right-sided tumors, anti-VEGF treatments were more effective. The results of these studies led to NCCN guideline recommendations based on tumor sidedness. For patients with metastatic left-sided KRAS wt tumors, cetuximab or panitumumab is recommended with chemotherapy. For patients with metastatic right-sided KRAS wt tumors, bevacizumab plus chemotherapy is recommended.

## 7. The HER2 Pathway

Human epidermal growth factor receptor 2 (HER2) is a receptor tyrosine kinase protein. The ERBB2 gene encodes HER2, which is a proto-oncogene that is implicated in a number of malignancies. Approximately 3–5% of colorectal cancers demonstrate HER2 amplification [68]. Recent investigations of multiple therapies targeting this pathway have demonstrated promising results among patients with HER2 amplification.

### 7.1. Tucatinib

Tucanitinib is an oral tyrosine kinase inhibitor that is HER2 selective. Tucanitinib has been investigated in combination with trastuzumab, another HER2 inhibitor [69]. In fact, the tucanitinib/trastuzumab combination was the topic of the MOUNTAINEER trial [70]. This phase 2 study was conducted in patients with disease progression on standard therapy. The objective response rate was 38.1% in patient’s refractory to standard therapy, leading to accelerated FDA approval for patients with the treatment of refractory mCRC. A phase 3 study, MOUNTAINEER-03, is underway to evaluate tucanitib/trastuzumab plus mFOLFOX versus mFOLFOX alone or in combination with bevacizumab or cetuximab (NCT0525351). 

### 7.2. Pertuzumab

Pertuzumab is a monoclonal antibody that targets HER2. Pertuzumab is often combined with trastuzumab for patients with HER2-positive breast cancer. In turn, the MyPathway trial investigated this combination in patients with HER2-amplified mCRC who had disease progression on standard therapy [71]. In 57 patients with HER2-amplified mCRC, there was an objective response rate of 32% with tolerable toxicity. Future larger clinical trials are underway to investigate the effectiveness of pertuzumab versus standard regimens.

### 7.3. Lapatinib

Lapatinib is a tyrosine kinase inhibitor that targets both HER2 and EGFR receptors. Lapatinib has been examined in combination with trastuzumab in the HERACLES trial [72]. This phase 2 trial evaluated the response and safety of the lapatinib/trastuzumab combination in 27 patients with KRAS wt and HER2+ tumors. At a median follow-up of 94 weeks, 30% of patients had achieved an objective response with no grade four or five events. Follow-up data demonstrated a median overall survival of 10.0 months, with one person achieving sustained complete response seven years after therapy [73].

### 7.4. Trasuzumab Deruxtecan

Trastuzumab deruxtecan is a drug conjugate consisting of the anti-HER2 monoclonal antibody, trastuzumab, and a topoisomerase I inhibitor, deruxtecan. The use of trastuzumab deruxtecan for patients with colorectal cancer was evaluated in the DESTINY-CRC01 multicenter phase 2 trial [74]. Patients with metastatic HER2+ CRC who had progressed on 2+ regimens were enrolled. Among 86 enrolled patients, 45.3% had an objective response. Median overall survival was 15.5 months with a reported 6.9-month progression-free survival. The phase 2 DESTINY-CRC02 trial is an ongoing study to assess for safety and efficacy of multiple doses (ClinicalTrial.gov ID: NCT04744831).

### 7.5. Summary

HER2 amplification is rare in patients with metastatic colorectal cancer, but multiple treatment options have been developed. The most promising results have been with dual HER2 targeted therapy. Current NCCN guidelines recommend trastuzumab + (pertuzumab, lapatinib, or tucatinib) or trastuzumab deruxtecan for patients with HER2 amplification and metastatic colorectal cancer.

## 8. Immune Checkpoint Inhibition

Approximately 15% of CRC tumors are characterized by microsatellite instability (MSI) [75]. These tumors have impaired DNA mismatch repair, leading to oncogenesis. Tumors with MSI tend to have a better prognosis but respond differently to standard chemotherapeutic regimens [75]. Several immunotherapies have been employed as checkpoint inhibitors to help improve antineoplastic immune response. The primary targets of these inhibitors are programmed cell death–ligand 1 (PD-L1), programmed cell death protein 1 (PD-1), and cytotoxic T-lymphocyte-associated protein 4 (CTLA-4). In patients with MSI high tumors, immune checkpoint inhibitors have demonstrated an improved immune response to tumors and patient outcomes.

### 8.1. Pembrolizumab

Pembrolizumab is a humanized antibody that targets the PD-1 receptor of lymphocytes, which has demonstrated efficacy in several solid tumors. The KEYNOTE-177 trial was a multicenter, randomized trial that evaluated pembrolizumab against chemotherapy (5-Fu based with or without bevacizumab or cetuximab) as first-line therapy in patients with MSI high tumors [76]. Pembrolizumab was associated with improved progression-free survival (median 16.5 months versus 8.2 months, HR 0.60, *p* < 0.001) and mean overall survival (13.7 months versus 10.8 months). Among patients with a response, treatment responses were durable at 24 months in 83% of patients treated with pembrolizumab versus 33.1% among patients who received standard chemotherapy. These data led to the first FDA approval of a single targeted agent as first-line therapy for mCRC.

### 8.2. Nivolumab and Ipilimumab

Dual checkpoint inhibition has been proposed as a therapeutic strategy in patients with MSI high tumors [77]. Nivolumab is a humanized monoclonal antibody targeted at PD-1. Ipilimumab, another monoclonal antibody, was developed to target CTLA-4. The CheckMate 142 trial was a multinational, multicenter, phase 2 trial that evaluated nivolumab in patients with MSI high tumors refractory to chemotherapy. Initial results demonstrated a 31% objective response rate and 69% demonstrated at least 12 weeks of stable disease [78]. Data from another cohort of the CheckMate 142 trial evaluated nivolumab plus ipilimumab in patients with MSI high tumors refractory to other treatments [79]. The objective response rate was 55%, and overall survival at 12 months was 85%. The most recent iteration of the CheckMate 142 trial evaluated nivolumab plus ipilimumab among patients with no prior treatments [80]. The objective response rate was 69%, and the disease control rate was 84%; there was a 13% complete response rate. Median progression-free survival and overall survival had not been met at 24.2 months. A phase 3 trial is ongoing to evaluate nivolumab, nivolumab/ipilimumab, and the investigator’s choice of chemotherapy (NCT04008030). 

### 8.3. Atezolizumab

Atezolizumab is a humanized monoclonal antibody that targets PD-L1, which was investigated in the IMblaze 370 trial [81]. This multinational, multicenter, phase 3 trial evaluated atezolizumab plus cobimetinib (MEK inhibitor) or atezolizumab monotherapy or regorafenib in patients with the treatment of refractory mCRC. MSI high tumors were capped at 5%. Atezolizumab failed to demonstrate improved survival in combination or as monotherapy versus regorafenib. The AtezoTRIBE trial added atezolizumab to FOLFOXIRI plus bevacizumab among patients with microsatellite stable tumors to assess the hypothesis that FOLFOXIRI and bevacizumab may increase the immunogenicity of microsatellite stable tumors [82]. The atezolizumab group had a 13.1-month progression-free survival versus 11.5 months in the control group with a roughly 30% decreased hazard ratio of death (HR = 0.69; *p* = 0.012). Multiple other clinical trials are currently investigating atezolizumab in both MSI high and microsatellite stable tumors.

### 8.4. Summary

Patients with mCRC and MSI respond differently to standard chemotherapy regimens. For patients with MSI high tumors, immune checkpoint inhibitors have shown promising results with impressive treatment response and significant improvements in overall survival. Either pembrolizumab or nivolumab/ipilimumab is recommended as first-line treatment for MSI high tumors. Atezolizumab is an emerging therapy, but additional data are needed prior to its approval for use in patients with mCRC.

## 9. Tyrosine Receptor Kinase Fusions

Neurotrophic tyrosine receptor kinase (NTRK) fusions are rare but targetable genetic mutations within CRC. These mutations are detected in <1% of CRC [83]. NTRK fusion leads to constitutive activation of TRK and subsequently to oncogenesis. Due to the rare nature of these tumors, clinical trials have evaluated efficacy across multiple gastrointestinal malignancies rather than specifically colorectal cancer. Two therapies, larotrectinib and entrectinib, are FDA-approved for use in CRC with NTRK mutation.

### 9.1. Entrectinib

Entrectinib was developed as an inhibitor of TRK and has been examined for multiple tumor types in phase 1 and 2 clinical trials [84]. At a median follow-up of 12.9 months, 57% of patients had an objective response and 7% of patients had a complete response; the median duration of response was 10 months. Among the fifty-four patients included in the study, four individuals had metastatic colorectal cancer. Subsequently, the FDA approved entrectinib for NTRK tumors.

### 9.2. Larotrectinib

Larotrectinib, a TRK inhibitor, has been tested in phase 1/2 studies for patients with TRK fusion-positive solid tumors [85]. In this trial, 8 of 159 patients had colon cancer; 4 patients with colon cancer had a response to therapy with a median duration of 3.7 months. Based on the efficacy across tumor types, larotrectinib was FDA-approved for tumors with NTRK fusions, including CRC.

### 9.3. Summary

NTRK fusions are extremely rare mutations in mCRC. The rarity of NTRK fusions has led to few clinical trials specifically focused on this patient population. Nevertheless, entrectinib and lartotrectinib have demonstrated activity in patients with solid tumors with NTRK fusion, and the FDA has approved their use for these patients.

## 10. Future Directions and Conclusions

The past two decades have been characterized by a tremendous increase in interest and investigation into targeted therapies for metastatic colorectal cancer. Multiple new promising therapies have gained FDA approval to treat metastatic colorectal cancer; in fact, subsets of patients have demonstrated complete and durable responses to targeted therapy. Nevertheless, mutations in KRAS, BRAF, and PTN can lead to resistance to certain targeted treatments, including EGFR therapies [86]. While anti-angiogenesis therapies have promise to prolong progression-free survival, drug resistance has also limited the number of patients who benefit relative to long-term overall survival [51]. Combination-targeted therapies may represent a mechanism to overcome resistance mechanisms [12]. In turn, the identification of mechanisms to improve immune response to cancer, including colorectal cancer vaccines and CAR-T cell therapy, is a topic of future interest [87,88]. In addition, new biomarkers are needed to identify which patients will benefit the most from targeted drugs [89]. Future studies should seek to improve outcomes for patients with metastatic colorectal cancer through a more personalized targeted approach.

## Figures and Tables

**Figure 1 cells-13-00245-f001:**
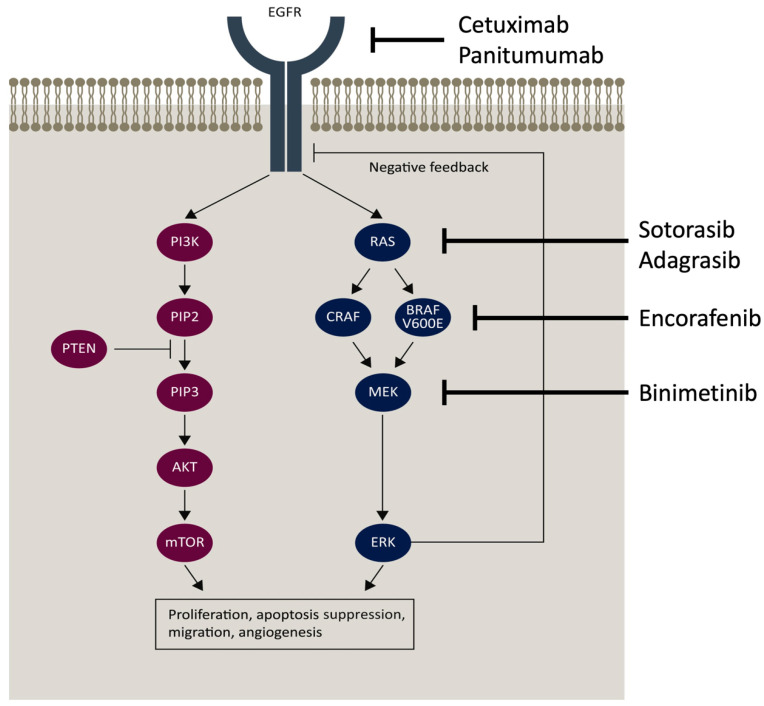
The EGFR pathway demonstrates multiple targets, including the EGFR, RAS, BRAF V600E, and MEK. The figure was adapted from [13].

**Figure 2 cells-13-00245-f002:**
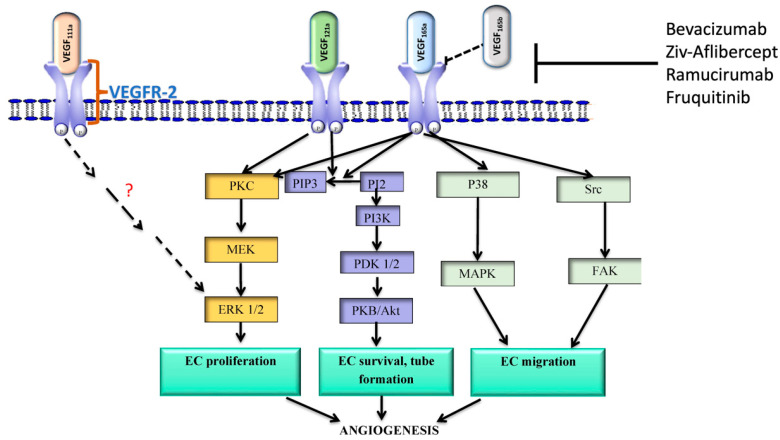
The VEGF pathway illustrates the mechanism through which the VEGF receptor elicits angiogenesis. The figure was adapted from [44].

**Table 1 cells-13-00245-t001:** FDA Approved targeted therapies for metastatic colorectal cancer.

Targeted Therapy	Trade Name	Year of Approval	Pathway
Cetuximab	Erbitux	2004	EGFR inhibitor
Panitumumab	Vectibix	2006	EGFR inhibitor
Bevacizumab	Avastin	2004	VEGF inhibitor
Ramucirumab	Cyramza	2015	VEGF inhibitor
Aflibercept	Zaltrap	2012	VEGF inhibitor
Fruquintinib	Fruzaqla	2023	VEGF inhibitor
Encorafenib	Braftovi	2020	BRAF inhibitor
Trastuzumab	Herceptin	2022	HER2 inhibitor
Pertuzumab	Perjeta	2022	HER2 inhibitor
Tucatinib	Tukysa	2023	HER2 inhibitor
Larotrectinib	Vitrakvi	2018	NTRK inhibitor
Entrectinib	Rozlytrek	2019	NTRK inhibitor
Regorafenib	Stivarga	2012	Multi-kinase inhibitor
Nivolumab	Opdivo	2017	PD-1 inhibitor
Pembrolizumab	Keytruda	2017	PD-1 inhibitor
Ipilimumab	Yervoy	2018	CTLA-4 inhibitor

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
