# Peer review of "Update on Targeted Therapy and Immunotherapy for Metastatic Colorectal Cancer"

_cells, 2024, doi:10.3390/cells13030245_

Round 1
Reviewer 1 Report
Comments and Suggestions for Authors
The review article entitled "update on targeted therapy for metastatic colorectal cancer" is primarily focused on summarizing the targeted therapies for metastatic colorectal cancer. Within a limitation of a review article, authors summarized the recent updates on using targeted therapies for treating metastatic colorectal cancer. Overall, the manuscript is well-written with an easier flow of reading. However, I would still suggest to edit the article once more to improve the flow of few sentences; for example, the use of word 'inform' does not read good in the lines 14 and 41, hence needs to be replaced with the word such as 'define', and the sentence should end with 'behavior and prognosis of the disease'.
Comments on the Quality of English LanguageOverall, the quality of english language is quite good, may need few minor edits to improve the flow of few sentences.
Author Response
The review article entitled "update on targeted therapy for metastatic colorectal cancer" is primarily focused on summarizing the targeted therapies for metastatic colorectal cancer. Within a limitation of a review article, authors summarized the recent updates on using targeted therapies for treating metastatic colorectal cancer. Overall, the manuscript is well-written with an easier flow of reading. However, I would still suggest to edit the article once more to improve the flow of few sentences; for example, the use of word 'inform' does not read good in the lines 14 and 41, hence needs to be replaced with the word such as 'define', and the sentence should end with 'behavior and prognosis of the disease'.
Thank you for your review of the article. We have edited the manuscript as requested to improve the flow of reading. We have specifically addressed the sentences noted by the reviewer.
Reviewer 2 Report
Comments and Suggestions for Authors
This manuscript focuses on targeted therapies in metastatic colorectal cancer. The authors collect recent trials targeting the EGF receptor, the VEGF, and the HER2 pathways, immune checkpoint inhibitors, tyrosine receptor kinase fusions. They also included data comparing EGFR and VEGF pathway inhibitions. The review is generally well written and easy to follow, and it focuses on a clinically important aspect of colorectal cancer. However, I have some comments that should be addressed by the authors.
1. For chapters 3,4, and 6, summary figures should be included to show the exact targeting points and/or mechanisms of the individual agents. This would largely visually help the reader to compare the effects of the therapies. In addition, summary tables to compare the effectiveness of the agents are also welcome.
2. The review is a list of agents and a list of survival numbers in its present form. Could the authors include summarizing remarks, and own interpretations at the end of each chapter? The readers are interested not only in the exact numbers, but in their meaning and interpretation, too.
3. Table 1: please correct "multikinase inhibtor" to "multikinase inhibitor".
4. Chapter 4.1: how does bevacizumab act? Could you please include a more detailed description on the mechanism of action of this agent?
5. Line 270: I guess "Fruquitinib.....to target the EGFR pathway" should be VEGFR pathway.
Author Response
This manuscript focuses on targeted therapies in metastatic colorectal cancer. The authors collect recent trials targeting the EGF receptor, the VEGF, and the HER2 pathways, immune checkpoint inhibitors, tyrosine receptor kinase fusions. They also included data comparing EGFR and VEGF pathway inhibitions. The review is generally well written and easy to follow, and it focuses on a clinically important aspect of colorectal cancer. However, I have some comments that should be addressed by the authors.
1. For chapters 3,4, and 6, summary figures should be included to show the exact targeting points and/or mechanisms of the individual agents. This would largely visually help the reader to compare the effects of the therapies. In addition, summary tables to compare the effectiveness of the agents are also welcome.
Thank you for this suggestion. We have added the targeting point to Figures 1 and 2 as requested. We did not create a new figure for HER2 as all the targets are the HER2 receptor, not a multiprotein pathway.
2. The review is a list of agents and a list of survival numbers in its present form. Could the authors include summarizing remarks, and own interpretations at the end of each chapter? The readers are interested not only in the exact numbers, but in their meaning and interpretation, too.
Thank you for this idea. We have added summarizing remarks at the end of the appropriate chapters.
3. Table 1: please correct "multikinase inhibtor" to "multikinase inhibitor".
This mistake has been corrected.
4. Chapter 4.1: how does bevacizumab act? Could you please include a more detailed description on the mechanism of action of this agent?
We added a more detailed description in line 241.
5. Line 270: I guess "Fruquitinib.....to target the EGFR pathway" should be VEGFR pathway.
Thank you. This mistake was corrected.
Reviewer 3 Report
Comments and Suggestions for Authors
In this review the authors summerize adequately the recent therapeutic regiments given to metastatic colorectal cancer patients as well as their effects. In addition they provide a glimpse to potentially future therapies.
No further comments by me!
Author Response
In this review the authors summerize adequately the recent therapeutic regiments given to metastatic colorectal cancer patients as well as their effects. In addition they provide a glimpse to potentially future therapies.
No further comments by me!
Thank you for your positive comments on our article.